# Strategies for optimising early detection and obstetric first response management of postpartum haemorrhage at caesarean birth: a modified Delphi-based international expert consensus

Verónica Pingray [1], Caitlin R Williams [1,2] Fadhlun M Alwy Al-beity,[3,4] Edgardo Abalos,[5,6] Sabaratnam Arulkumaran,[7] Alejandro Blumenfeld,[1,8] Brendan Carvalho [9] Catherine Deneux-Tharaux,[10] Soo Downe,[11,12] Alexandre Dumont,[13] Maria Fernanda Escobar,[14,15] Cherrie Evans,[16] Sue Fawcus,[17] Hadiza S Galadanci,[18,19] Diem-Tuyet Thi Hoang,[20] G Justus Hofmeyr,[21,22] Caroline Homer,[23] Ayodele G Lewis,[24] Tippawan Liabsuetrakul [25,26] Pisake Lumbiganon,[27] Elliott K Main,[28,29] Judith Maua,[30] Francis G Muriithi [31,32] Ashraf Fawzy Nabhan [33] Inês Nunes,[34,35,36] Vanesa Ortega,[37] Thuan N Q Phan [38,39] Zahida P Qureshi,[40] Claudio Sosa,[41,42] John Varallo,[43] Andrew D Weeks,[38,44] Mariana Widmer,[45] Olufemi T Oladapo,[45] Ioannis Gallos,[45] Arri Coomarasamy,[31] Suellen Miller,[46] Fernando Althabe[45]

For numbered affiliations see end of article.

**Correspondence to**
Verónica Pingray;
vpingray@iecs.org.ar

## ABSTRACT

**Objective** There are no globally agreed on strategies on early detection and first response management of postpartum haemorrhage (PPH) during and after caesarean birth. Our study aimed to develop an international expert's consensus on evidence-based approaches for early detection and obstetric first response management of PPH intraoperatively and postoperatively in caesarean birth.

**Design** Systematic review and three-stage modified Delphi expert consensus.

**Setting** International.

**Population** Panel of 22 global experts in PPH with diverse backgrounds, and gender, professional and geographic balance.

**Outcome measures** Agreement or disagreement on strategies for early detection and first response management of PPH at caesarean birth.

**Results** Experts agreed that the same PPH definition should apply to both vaginal and caesarean birth. For the intraoperative phase, the experts agreed that early detection should be accomplished via quantitative blood loss measurement, complemented by monitoring the woman's haemodynamic status; and that first response should be triggered once the woman loses at least 500 mL of blood with continued bleeding or when she exhibits clinical signs of haemodynamic instability, whichever occurs first. For the first response, experts agreed on immediate administration of uterotonics and tranexamic acid, examination to determine aetiology and rapid initiation of cause-specific responses. In the postoperative phase, the experts agreed that caesarean birth-related PPH should be detected primarily via frequently monitoring the woman's haemodynamic status and clinical signs and symptoms of internal bleeding, supplemented by cumulative blood loss assessment performed quantitatively or by visual estimation. Postoperative first response was determined to require an individualised approach.

**Conclusion** These agreed on proposed approaches could help improve the detection of PPH in the intraoperative and postoperative phases of caesarean birth and the first

## STRENGTHS AND LIMITATIONS OF THIS STUDY

⇒ Use of a rigorous and systematic process to identify and synthesise high-quality postpartum haemorrhage (PPH) evidence in the literature.

⇒ The selection of the expert panellists ensured a wide range of perspectives to enhance the utility and applicability of this consensus to a wide range of clinical settings.

⇒ There was a very low rate of loss to follow-up and the first two rounds of the modified Delphi process were blinded to avoid social acceptability bias, and the hybrid meeting was facilitated to ensure that all panellists had equal opportunity to contribute to the discussion.

⇒ Due to the dearth of quality evidence on PPH related to caesarean birth, experts often had to extrapolate from evidence on interventions recommended for PPH in vaginal birth or make decisions based on their experiences.

⇒ Given the highly technical content, we did not include recipients of these interventions, or their representatives, among the panellists.

response management of intraoperative PPH. Determining how best to implement these strategies is a critical next step.

## INTRODUCTION

Deaths from postpartum haemorrhage (PPH), the leading direct cause of maternal mortality globally, are potentially preventable with timely diagnosis and management.[1 2] The risk of PPH is significantly higher with caesarean birth than vaginal birth, especially in cases of emergency caesarean birth.[3] With global caesarean birth rates increasing, PPH during and after caesarean birth is a growing concern.[4] The impact is particularly acute in low-income and middle-income countries (LMICs), where 32% of all maternal deaths after caesarean birth are related to PPH.[5] In some LMICs, caesarean births outnumber vaginal births.[6] Several factors challenge effective response to PPH in LMICs. These countries have well-documented difficulties accessing surgical services, skilled staff and blood/blood products.[7] Even when access concerns are addressed, the use of interventions to detect and manage PPH is often inconsistent.[8 9]

A standardised approach to PPH management has been shown to improve outcomes, including significantly reducing severe PPH rates among women giving birth vaginally.[10] Similarly, studies including women having caesarean birth suggest a reduction in severe morbidity associated with the use of comprehensive haemorrhage protocols.[11 12] WHO has published and updated recommendations for the prevention and treatment of PPH.[2 13 14] However, these recommendations neither detail methods for early detection of PPH during and after caesarean birth nor clearly indicate when to initiate treatment (ie, the 'trigger' criteria), both of which may contribute to observed variations in clinical practice.[2 7 15] PPH management practices may vary depending on whether the haemorrhage occurs during or after the surgical procedure.[16] Proposing standardised and evidence-based global strategies may help to reduce practice variations and improve the quality of care. Our study aimed to develop an international consensus on standardised approaches for PPH detection and obstetric first response management for women who develop primary PPH during and after caesarean birth, and at the time of initiating treatment, the suspected aetiology is uterine atony, traumatic PPH or unknown.

## METHODS

The study involved a systematic review and an expert consensus using a three-stage modified Delphi process.

### Systematic review

A systematic review of published national and international guidelines for PPH prevention and management was conducted to identify interventions for collecting and measuring blood loss, methods for detecting PPH, thresholds for treatment and first response conservative obstetric interventions to manage PPH both during

surgery (intraoperative) and after surgery (postoperative). The evidence summarised involved treatments options for women who develop primary PPH during or after caesarean birth, and at the time of initiating treatment, the suspected aetiology is either uterine atony, traumatic PPH or unknown. Treatments for managing women with a diagnosis of antepartum haemorrhage, coagulopathy, placenta previa or placenta accreta were not included, given that treatments are usually specific to each aetiology. To be included, the guidelines needed to include guidance on the detection or management of PPH during or after caesarean birth. The literature search in PubMed, EMBASE, CINAHL and Cochrane Library databases included papers published from January 2012 to July 2022 (online supplemental file 1). The search was complemented by reviewing the English-language grey literature to identify guidelines.

Since few of these guidelines were focused specifically on the intraoperative or postoperative phases or described PPH detection methods, an additional systematic search was conducted, focused on PPH detection and conservative obstetric first response management during and after caesarean birth. Peer-reviewed systematic reviews of randomised controlled trials (RCTs) were eligible. Subject matter experts were consulted to add any relevant peer-reviewed articles missed by the systematic search.

Titles and abstracts of both guidelines and systematic reviews of RCTs were screened by pairs of independent reviewers who subsequently reviewed full texts, conducted quality appraisals and extracted data using previously piloted forms. Only guidelines with AGREE II (Appraisal of Guidelines for Research & Evaluation) scores between 5 and 7 and systematic reviews with modified AMSTAR (A MeaSurement Tool to Assess systematic Reviews) quality assessment of 'moderate' or 'high' were eligible for data extraction.[17 18] The results of the systematic review were used to inform the development of the Delphi surveys and to provide the experts with summaries of the existing evidence. Additional methodological details can be found in the online supplemental file 2.

### Expert consensus

A three-stage modified Delphi process was conducted between December 2021 and September 2022, with two rounds of individual online surveys, followed by a third round: a hybrid (virtual and in-person) meeting with group discussions and final voting. Twenty-five PPH experts with the knowledge and skills to critically assess scientific evidence were invited to participate in all three rounds. They included specialists in nursing, midwifery, obstetrics, surgery and anaesthesia. The experts were selected to ensure gender, professional and geographic balance. Most experts were coauthors of recent national and international guidelines or principal or co-investigators of PPH clinical trials. The same experts were invited to participate in all three rounds. In the third round, observers representing professional associations

**Table 1** Themes explored and criteria used to guide assessments

| Themes | Criteria and items included in each questionnaire |
|---|---|
| PPH definitions | ► Appropriateness of using a single definition for PPH, regardless of mode of birth<br>► Timeframe for postoperative PPH |
| Early detection methods<br>Intraoperative and postoperative | *Criteria*: clinical usefulness, feasibility of use in all settings attending caesarean birth, acceptability to key stakeholders and estimate of resources required<br>*Items*:<br>► Visual estimation of blood loss<br>► Volumetric assessment of blood loss<br>► Gravimetric assessment of blood loss<br>► Clinical signs of haemodynamic instability<br>► Visual charts and early warning scores<br>► Clinical judgement (eg, rate of flow, duration)<br>► Volumetric+gravimetric assessment of blood loss<br>► Volumetric/Gravimetric assessment of blood loss+clinical signs of haemodynamic instability<br>► Visual estimation+visual charts/early warning systems |
| Thresholds for action<br>Intraoperative and postoperative | *Criteria*: accuracy, feasibility of use in all settings attending caesarean birth and acceptability to key stakeholders<br>*Items*:<br>► *One-step approach* (single threshold triggers full response protocol)<br> – At least 500 mL blood loss alone<br> – At least 1000 mL blood loss alone<br> – Haemodynamic instability alone<br> – At least 500 mL blood loss OR signs of haemodynamic instability<br> – At least 1000 mL blood loss OR signs of haemodynamic instability<br>► *Two-step approach* (lower threshold triggers further assessment, preparedness and close monitoring; higher threshold triggers initiation of treatment)<br> – Lower threshold of at least 500 mL, and higher threshold of at least 1000 mL blood loss OR signs of haemodynamic instability<br> – Lower threshold of at least 1000 mL, and higher threshold of at least 2000 mL blood loss OR signs of haemodynamic instability |
| First response conservative obstetric interventions<br>Intraoperative and postoperative | *Criteria*: balance of effects, feasibility of use in all settings attending caesarean birth, acceptability to key stakeholders, estimate of resources required, equity<br>*Items*:<br>► Oxytocin<br>► Carbetocin<br>► Tranexamic acid<br>► Compressive sutures<br>► Bimanual compression<br>► Uterine massage<br>► Oxytocin-ergometrine fixed dose<br>► Prostaglandin<br>► Ergometrine<br>► Non-pneumatic antishock garment<br>► External aortic compression<br>► Intrauterine balloon tamponade |

PPH, postpartum haemorrhage.

and WHO regional offices, or who were leaders in PPH research were invited to share their views, but were not eligible to vote.

Based on the findings of the systematic review, questionnaires with open-ended and close-ended questions were developed, piloted and administered using Survey Monkey. A summary of the themes and interventions included in the surveys and criteria used to guide judgements are described in table 1. The criteria, methods, interventions and other items included in the surveys were presented with definitions to facilitate interpretation. The themes were explored separately for the intraoperative and postoperative phases. Experts had to consider PPH detection methods and first response obstetric interventions to be applied in any type of Comprehensive Emergency Obstetric and Newborn Care services facility, and applicable for primary PPH. In line with the scope of the systematic review, the consensus targeted conservative first-response obstetric interventions applicable for women with any cause of PPH until the main cause of

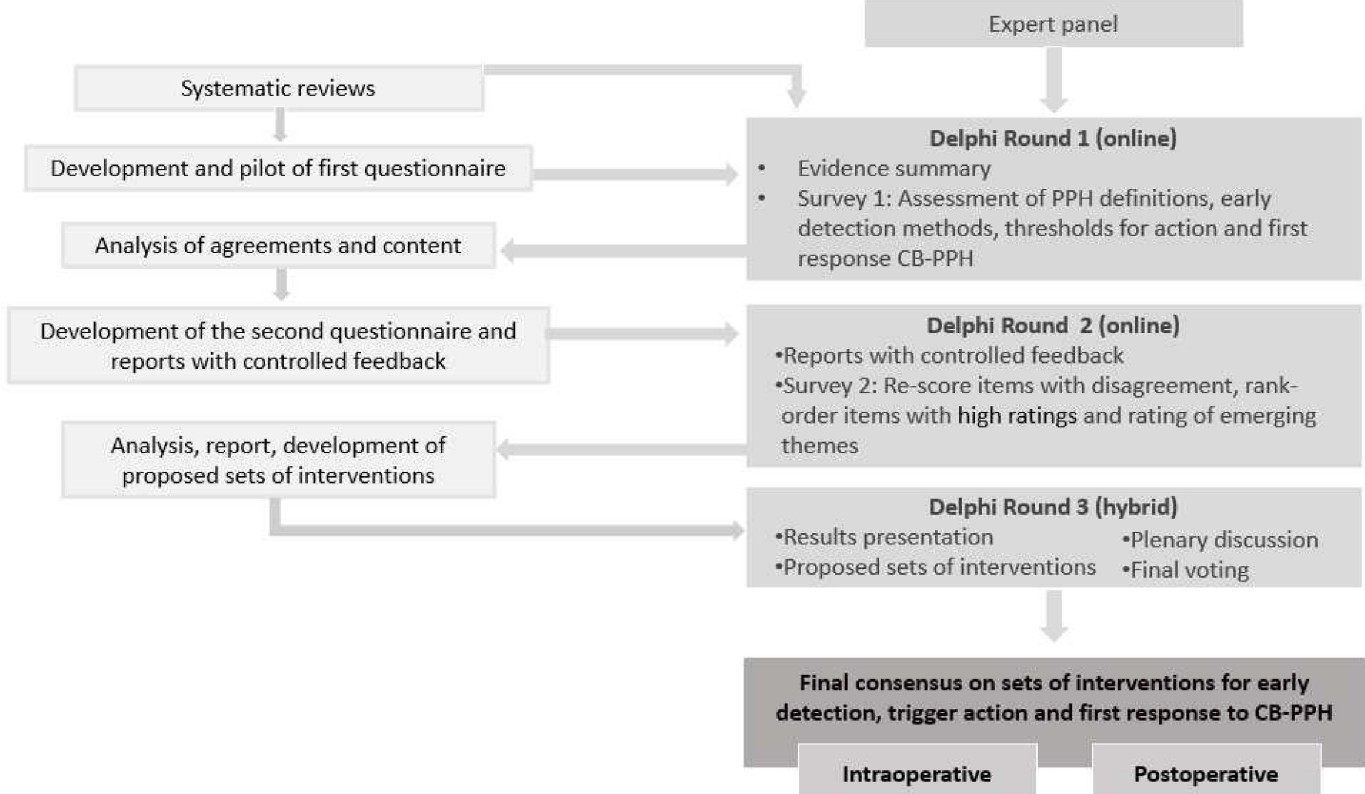

**Figure 1** Technical consultation flow chart. CB, caesarean birth; PPH, postpartum haemorrhage.

PPH is identified, and diagnosis of atonic and traumatic PPH. It did not target the first response to women with diagnosis of PPH due to placenta previa, placenta accreta, coagulopathies and retained tissue. Although most cases of PPH are controlled by the simultaneous application of obstetric interventions and haemostatic support,[19] this consensus focused mainly on obstetric interventions and not haemostatic resuscitation and treatment of anaemia and coagulopathy.

Experts were asked to consider the postoperative PPH phase as only the first 2 hours immediately after the operation. Each online survey was available for response for 6 weeks, and three reminders were sent to participants with incomplete or no responses. In the first round, experts were asked to rate caesarean-related PPH definitions, detection methods, thresholds to trigger treatments and first response conservative obstetric interventions. In the second round, experts received their previous individual ratings and group rating distributions. They were asked to re-rate detection methods with disagreement, rank-order the thresholds and first response treatments that had previously received high ratings and rate new questions that emerged from experts' comments in open-ended questions from round 1. In the third round, experts met for a 2-day hybrid meeting to discuss areas of divergence between surveys' findings and to rate (anonymously) the final sets of interventions. The agenda and questions guide used to facilitate the discussion are available in the online supplemental file 3,4. Figure 1 outlines the process of consensus building.

Median group rating and disagreement index (DI) were calculated to summarise experts' ratings and to measure agreement. A DI <1 indicated agreement, while a DI ≥1 indicated disagreement.[20] The RAND/UCLA (Research and Development Organization/University of California at Los Angeles) appropriateness scale was used to classify interventions as 'appropriate', 'inappropriate' or 'uncertain'.[20] Interventions with median ratings in the top third of the appropriateness scale[7–9] were classified as 'appropriate'; those in the bottom third were classified as 'inappropriate'[1–3] and those with intermediate median ratings were classified as 'uncertain'.[4–6] Domains with disagreement among the experts were also classified as 'uncertain' (online supplemental figure 1).

### Patient and public involvement
Given the highly technical content, we did not include recipients of these interventions, or their representatives, among the panellists. This limitation is addressed in the 'Discussion' section.

### RESULTS
The systematic search identified 802 guidelines and systematic reviews. After screening and quality appraisal, 17 guidelines,[2 13 15 21–34] 4 systematic reviews[35–38] and 15 peer-reviewed studies[39–54] were included (online supplemental figure 2). Included guidelines and systematic reviews identified 6 PPH definitions, 5 PPH detection methods, 10 blood loss collection devices, 7 thresholds

to initiate treatment and 14 obstetric interventions to manage PPH conservatively. Results are in online supplemental tables 1–4.

Of the 25 experts invited, 22 agreed to participate in the Delphi process (online supplemental file 5). All completed the first and second rounds, while 20/22 participated and voted in the third round. The experts who completed all rounds were from 11 countries from all WHO world regions (6 from the African Region, 1 from the Eastern Mediterranean Region, 3 from the European Region, 6 from the Region of the Americas, 2 from the South-East Asian Region and 2 from the Western Pacific Region). They had different professional backgrounds (obstetricians and gynaecologists, anaesthetists, surgeons, nurse-midwives and midwives) and were gender-balanced (12 men and 10 women). In addition, four observers participated in the discussion during the third round but did not vote.

The median ratings and measures of agreement obtained from the first and second rounds of online surveys are given in online supplemental tables 5–8 and online supplemental figure 3. Experts' ratings and agreements in the third round are given in table 2. Consensus was reached for (a) using a single definition for PPH, regardless of mode of birth, (b) early detection of PPH at caesarean birth and thresholds to initiate treatment in the intraoperative phase, (c) clinical interventions for conservative obstetric first response management of intraoperative PPH and (d) early detection of PPH after caesarean birth and thresholds to initiate treatment in the postoperative phase. However, the first response treatment in the postoperative phase was determined to require an individualised approach.

### Definition of PPH during and after caesarean birth
The experts agreed that a single definition of PPH should be used, regardless of mode of birth (median rating 7.5; DI −5.23). Specifically, they agreed that the definition of PPH during and after caesarean birth should be the same as the definition of PPH related to vaginal birth, to underscore the importance of rapid action to address excessive bleeding.

### Intraoperative phase
#### Early detection of PPH during caesarean birth and thresholds for triggering action
Experts agreed that during caesarean birth, blood loss should be assessed via quantitative measurement, complemented by ongoing monitoring of the woman's haemodynamic status (median rating 8; DI −0.34). Furthermore, quantitative measurement and monitoring should be incorporated into routine practice alongside strategies to prevent PPH (box 1). They noted the importance of distinguishing blood from amniotic fluid. This might be achieved by using separate suction canisters or measuring and recording the amount of amniotic fluid within the canister immediately after birth and before delivery of the placenta. Some experts noted that the assessment of atonic PPH may require installing and monitoring under-buttock drapes to assess vaginal blood loss.

The experts agreed that first response treatment should be triggered if the woman has lost at least 500 mL of blood and still has continued bleeding or if she exhibits clinical signs of haemodynamic instability, whichever occurs first (median rating 8; DI −0.13). Such early action was considered important to prevent severe PPH and associated morbidity, because measurement of blood loss lags actual blood loss. Rapid response has been identified as a critical component of the effectiveness of an early detection and PPH treatment strategy to prevent severe PPH in vaginal births.[40] Experts considered that rapid response is particularly important in settings with a high prevalence of anaemia. It was noted that the proposed threshold for triggering action may result in many women receiving first response treatment for PPH. Some experts pointed out that this could diminish providers' responsiveness and recognition of PPH as a serious complication.

Several experts flagged the need for guidance in determining when haemodynamic instability occurs. They noted that healthcare providers are often diligent in recording vital signs, but may not know when to escalate care. Although beyond the scope of this consensus, the provision of guidance to clinicians was acknowledged.

### First response management: intraoperative phase
The agreed first response management is summarised in box 2. Specifically, the experts agreed clinicians should commence an infusion of oxytocin. If a prophylactic or other oxytocin infusion is already in place, the anaesthetist should quickly maximise the oxytocin dose as increasing uterine tone helps to reduce bleeding from the incision. If atony is diagnosed or the bleeding continues, the anaesthetist should rapidly add in a different uterotonic for treatment. The experts noted that this should occur quickly, rather than waiting to see whether the bleeding is responsive to oxytocin. They also agreed that tranexamic acid (TXA) should be administered as first response treatment, unless the woman had already received TXA within the last 30 min. Next, the team should carefully examine the woman to determine the source(s) of bleeding and initiate a cause-specific response. If the bleeding is due to trauma, the surgical team should close the uterus, repair any tears and attend to the wound. If the bleeding is due to uterine atony, the surgical team should control bleeding mechanically with intra-abdominal uterine massage or massage the exteriorised uterus, as the anaesthetic team manages uterotonic administration, as previously described. Experts highlighted that bleeding may be due to a combination of trauma and uterine atony; in such cases, the team should take a comprehensive approach. The experts also highlighted the importance of exteriorising and examining the posterior side of the uterus for tears and occult uterine rupture.

Surgical and anaesthetic teams should mobilise to administer the surgical and medical first responses concurrently. Team communication can be challenging

**Table 2** Experts' ratings and agreement on early detection and thresholds for triggering action for the first response to intraoperative and postoperative PPH

| Set of clinical interventions | Rating distribution | | | Agreement | | | Appropriateness scale‡ |
|---|---|---|---|---|---|---|---|
| | # votes in each interval | | | Median (IQR) in a 1–9 scale | RAND DI* | Qualitative scale† | |
| | 1–3 | 4–6 | 7–9 | | | | |
| **Intraoperative** | | | | | | | |
| Early detection of PPH in all women having caesarean birth | | | | | | | |
| Quantitative blood loss measurement and monitoring of haemodynamic status | 0 | 1 | 19 | 8 (1) | −0.34 | Yes | Appropriate |
| Thresholds for triggering action | | | | | | | |
| Option 1: at least 500 mL with continued bleeding OR clinical signs of haemodynamic instability | 1 | 3 | 16 | 8 (1) | −0.13 | Yes | Appropriate |
| Option 2: at least 750 mL with continued bleeding OR clinical signs of haemodynamic instability | 7 | 4 | 9 | 6 (5) | 3.13 | No | Uncertain |
| **Postoperative** | | | | | | | |
| Early detection of PPH in all women having caesarean birth | | | | | | | |
| Monitoring of haemodynamic status and quantitative blood loss measurement if feasible | 0 | 2 | 18 | 8 (1) | −0.34 | Yes | Appropriate |
| Thresholds for triggering action | Not applicable | | | | | | |

*RAND DI: the disagreement index is a continuous scale used to measure the dispersion of experts' ratings, taken as an indicator of the level of agreement.
†Agreement: a DI <1 represents an agreement, while a DI ≥1 indicates disagreement.
‡Appropriateness: items are classified as 'appropriate' with median ratings in the top (median between 7 and 9) third and agreement, and as 'uncertain' with intermediate median ratings (median between 4 and 6) or with disagreement.
DI, disagreement index; PPH, postpartum haemorrhage.

---

**Box 1    Agreed early detection of postpartum haemorrhage (PPH) during caesarean birth and thresholds for triggering first response in the intraoperative phase**

Early detection of PPH during caesarean birth
⇒ Quantitative measurement of blood loss
  ⇒ Volumetric measurement alone if feasible (able to capture all blood)
  ⇒ Volumetric measurement+gravimetric measurement
⇒ Monitor haemodynamic status
Thresholds for triggering first response
⇒ At least 500 mL measured blood loss WITH continued active bleeding OR
⇒ Clinical signs of haemodynamic instability

**Additional comments**
It is important to separate/distinguish amniotic fluid from blood.
All blood loss may not be immediately obvious. Examine the posterior side of the uterus for cervical tears and occult uterine rupture, and install and monitor an underbuttock drape to assess vaginal blood loss.
To prevent severe PPH, first response management should be triggered early if there is still continued bleeding, particularly in settings with a high prevalence of anaemia or where unavoidable delays implementing treatment are anticipated.

---

**Box 2    Agreed on first response treatment for postpartum haemorrhage (PPH) during the intraoperative phase**

⇒ At least 500 mL measured blood loss WITH ongoing bleeding OR clinical signs of haemodynamic instability:
⇒ If already infusing oxytocin, maximise dose OR add alternative uterotonic. If not already infusing, commence oxytocin infusion.
⇒ Tranexamic acid (TXA) (1 g in 10 mL intravenous over 10 min), if not already administered within the last 30 min.
⇒ Examine and rapidly initiate cause-specific response:
  ⇒ If from incision or surgical trauma: rapid haemostasis: close uterus, repair tears, attend to the wound.
  ⇒ If atony/placental cause: uterotonics (as above) and control bleeding mechanically with intra-abdominal uterine massage or exteriorise the uterus and massage.

**Additional comments**
Medical and surgical first responses should be administered concurrently, and effective team communication is key.
Replace intravenous fluids as needed for haemodynamic maintenance, according to the clinical condition, estimated blood loss and local protocols.
TXA should be administered as first response treatment, unless the woman has already received TXA for PPH prevention or treatment within the last 30 min. Up to two doses of TXA, at least 30 min apart may be administered.
If atony is diagnosed or the bleeding continues after the oxytocin dose has been maximised, the anaesthetists should rapidly add in a different uterotonic for treatment.
Due to quantity and rapidity of blood loss, there may be some cases that require an individualised approach.

---

and should be practised in drills to develop effective messages that will not alarm women. Teams should immediately call for senior assistance when necessary.

Experts also noted that anaesthetic teams should replace fluids as needed for haemodynamic maintenance, according to the clinical condition and estimated blood loss. Some experts noted that providing guidance on amounts of fluids was too case-specific. However, others stressed that inexperienced clinicians needed concrete guidance to avoid adding too many fluids and inducing fluid overload. Although this type of guidance is beyond the scope of this study, it is a relevant issue that should be addressed.

Experts stressed the importance of ensuring adequate intravenous access (via a wide-bore cannula or a second cannula) early on to enable escalation, given that it can be difficult to establish as a woman loses greater blood volume.

Although outside the scope of this consensus, one expert noted that clinicians need to be thinking about coagulopathy: both as a possible cause of bleeding and as a side effect of resuscitation efforts, and that it requires specific guidance on appropriate blood products in all settings.

Finally, experts acknowledged that this first response approach is intended to be appropriate for most cases of intraoperative PPH. There may be some cases that, due to quantity and rapidity of blood loss, require an individualised approach. Placental aetiologies, such as placenta previa or accreta, may require specific first response surgical (eg, lower uterine compression sutures) or mechanical (eg, internal aortic compression) procedures. While these aetiologies were not specifically targeted

in this Delphi process, some first response actions were suggested by experts.

### Postoperative phase
#### Early detection of PPH after caesarean birth and thresholds for triggering action

Postoperative detection of PPH based on monitoring blood loss can be misleading because of internal bleeding. Thus, during this phase, experts agreed (median rating 8, DI −0.34) that blood loss should be assessed primarily through frequent monitoring of women's haemodynamic status (when possible, at least every 15 min for the first 2 hours) and clinical signs and symptoms of internal bleeding (eg, assessment of fundal height) (box 3). In addition, if the assessment of postoperative vaginal blood loss is feasible, either by quantitative measurement or estimation (eg, counting and weighting pads), it should be performed. Here, the experts noted that clinical teams should not rely on vital signs alone, as vital signs' disturbances can lag behind other clinical indications of haemorrhage. Some experts noted that postoperative monitoring for at least 30 min after caesarean birth should occur in a designated recovery area to ensure the woman's safety. If internal bleeding is suspected, experts recommended an urgent ultrasound assessment if available. Experts agreed that, when possible, measured postoperative blood loss should be added to the quantified intraoperative blood loss, although they acknowledged

> **Box 3  Agreed early detection of postoperative postpartum haemorrhage (PPH) and thresholds for triggering first response**
>
> **Early detection of PPH**
> ⇒ Frequent monitoring of haemodynamic status (at least every 15 min for the first 2 hours)
>    ⇒ Heart rate
>    ⇒ Blood pressure
>    ⇒ Shock index
>    ⇒ Clinical signs/symptoms suspicious of internal bleeding
> ⇒ Quantitative blood loss assessment, if feasible
>    ⇒ Measured or estimated postoperative blood loss (when possible, added to quantified intraoperative blood loss)
> **Thresholds for triggering first response management**
> ⇒ Clinical signs and symptoms of haemodynamic instability, in accordance with local protocols
>
> **Additional comments**
> Relying on postoperative blood loss alone can underestimate internal bleeding. Increase vigilance and assess haemodynamic status frequently.
> Early detection of postoperative PPH should mainly rely on frequent monitoring of haemodynamic status and clinical signs and symptoms of internal bleeding. If assessment of postoperative vaginal blood loss is feasible, either by quantitative measurement or estimation (eg, counting pads), it should be performed.
> When possible, assessed postoperative blood loss should be added to the quantified intraoperative blood loss.
> The cumulative intraoperative and postoperative blood loss, together with a woman's haemodynamic status, may better determine the frequency and characteristics of postoperative monitoring and thresholds for action.
> Haemodynamic parameter thresholds for vital signs and Obstetric Shock Index to trigger treatment are not yet agreed on.

that this may be challenging in some settings. Experts noted that cumulative intraoperative and postoperative blood loss, together with a woman's haemodynamic status, can help adjust the frequency and characteristics of postoperative monitoring and thresholds for action. For example, a woman who experienced substantial blood loss intraoperatively may require more frequent monitoring than the baseline every 15 min.

Although it is beyond the scope of this study, the experts acknowledged that providing guidance on haemodynamic parameters cut-off points for postoperative thresholds to trigger treatment will help clinicians act more quickly. Several experts raised the possibility of using the Obstetric Shock Index (OSI) (heart rate divided by systolic blood pressure; OSI) as a clinical decision support tool to simplify the decision of when to act, given that it has been used in some settings, including low-resource settings.

### Postoperative phase: first response management

The experts noted that the follow-on postoperative treatment approach may vary substantially according to many factors, including the woman's baseline risk, anaemia, whether intraoperative PPH occurred, the woman's

postoperative haemodynamic status and clinical signs and symptoms of internal bleeding (eg, assessment of fundal height; if available, ultrasound, paracentesis). Until further evidence is available, experts recommended that local protocols be developed that consider these factors, rather than relying on a common postoperative first response approach for all cases and settings.

### Experts' final comments

The experts recognised that detection methods and first response interventions for PPH are essential for the care of all women having a caesarean birth, regardless of their risk status. However, women at high risk of developing PPH may require additional specialised monitoring and care.

In addition, given that PPH can arise intraoperatively or postoperatively for any woman, strategies for early detection of PPH should be incorporated into routine practice alongside PPH prevention and risk assessment.

Finally, experts highlighted two cross-cutting remarks regarding PPH during and after caesarean birth. First, good surgical practices, as recommended by the WHO Guidelines for Safe Surgery, should be followed to prepare for, perform and follow-up caesarean births.[55] The routine use of WHO surgical safety checklists has proven beneficial in reducing perioperative complications[56] (see online supplemental file 6 for more details). Second, it was noted that teamwork, communication and cooperation are critical. Effectively implementing the early detection and first response interventions described will require training, supportive supervision, monitoring and evaluation.

### DISCUSSION
#### Main findings

Expert consensus on optimal approaches for detecting and managing PPH during and after caesarean birth was developed among an international panel. Through two systematic reviews and a three-round modified Delphi process, consensus was reached for (a) using a single definition for PPH, regardless of the mode of birth, (b) early detection of PPH during caesarean birth and thresholds to initiate treatment in the intraoperative phase, (c) clinical interventions for first response to intraoperative PPH and (d) early detection of PPH after caesarean birth and threshold to initiate treatment in the postoperative phase. First response treatment in the postoperative phase was determined to require an individualised approach.

#### Strengths and limitations

Study strengths include the use of a rigorous and systematic process to identify and synthesise PPH evidence in the literature. We conducted rigorous systematic reviews with detailed quality appraisals to ensure that we used only high-quality evidence to identify approaches for PPH detection and management interventions. The selection of the expert panellists ensured a wide range

of perspectives, to enhance the utility and applicability of this consensus to a wide range of clinical settings. There was a low rate of loss to follow-up. The first two rounds of the modified Delphi process were blinded, to avoid social acceptability bias, and the hybrid meeting was facilitated by members of the Steering Group, to ensure that all panellists had equal opportunity to contribute to the discussion. The staged modified Delphi process allowed ample time for discussion and input, and experts provided additional comments to refine the final statements for clarity and accuracy.

Limitations included a dearth of quality evidence on PPH related to caesarean birth. Despite ample evidence on PPH during and after vaginal birth, there is far less published evidence on caesarean birth. Often, the experts had to extrapolate from evidence on interventions recommended for PPH in vaginal birth and make decisions based on their experiences, expert opinions and best practices, rather than evidence from comparative research. In some cases, this led to omitting interventions that might be useful for early detection or first response management because there was no rigorous evidence available. It is also a limitation that, given the highly technical content, we did not include recipients of these interventions, or their representatives, among the panellists.

Additionally, since this systematic review of guidelines was conducted, three updated PPH guidelines have been published.[57–59] None of these guidelines are specific to PPH at caesarean birth, although all contain some guidance relevant to PPH during or after caesarean birth. The recommendations within these guidelines generally align with previously published guidance included in our study, with a few notable exceptions. The revised 2023 FIGO (International Federation of Gynecology and Obstetrics) PPH guideline recommends the use of the OSI (with a threshold of ≥0.9 triggering first response treatment), together with the rule of 30, while acknowledging that 'the association between shock parameters and advanced treatment modalities in severe PPH has yet to be reported'.[60] In the updated CMQCC (California Maternal Quality Care Collaborative) Obstetric Haemorrhage Toolkit, greater emphasis is placed on assessing for concealed haemorrhage. The guideline recommends using a combination of clinical signs of hypovolemia, the shock index and Early Warning Score to enable earlier postoperative PPH detection.[58] The Royal College of Physicians of Ireland guideline suggests that prophylactic TXA administration be considered in women at high PPH risk.[59] The timing of our study prevented us from incorporating these revisions into our systematic review.

## Interpretation

This expert consensus aligns with the recent expert consensus developed by the African Perioperative Research Group (APORG) Caesarean Delivery Haemorrhage Group[61] for clinicians working in Africa. The APORG expert consensus had a broader scope, encompassing antenatal and perioperative prevention, preparedness, first response, refractory treatment interventions, and community-level and health system-level indirect interventions. This present expert consensus focuses only on early detection and first response, including specific thresholds for triggering action.

With rates of caesarean birth rising globally, particularly in middle-income countries,[6] this research is timely and crucial. International initiatives are underway to end preventable deaths due to PPH, such as the Roadmap to Combat Postpartum Haemorrhage between 2023 and 2030,[62] and the Pan American Health Organization's Zero Maternal Deaths by Hemorrhage campaign.[63] The present expert consensus on early detection and first-response treatment for PPH at caesarean birth adds to existing efforts by clearly delineating how interventions need to be tailored for the context of caesarean birth. This consultation represents an important first step towards developing standardised strategies for reducing morbidity and mortality related to PPH during and after caesarean birth. Determining how best to implement these standardised strategies is a critical next step.

Insights from implementation science suggest that defining evidence-based interventions is a necessary but insufficient step towards changing clinical practice.[64] Establishing implementation approaches is believed to increase uptake and fidelity of evidence-based interventions.[65] Clinical bundles are one implementation approach that has gained traction in recent years.[57 66–69] Global evidence suggests that clinical bundles are a powerful implementation approach for early detection and first response for PPH after vaginal birth.[68 69] However, it is unclear whether a bundle is the most appropriate implementation approach for PPH during and after caesarean birth. Bundles require a set of interventions to be administered together, but the administration of some of the clinical interventions outlined here may depend on what occurs during surgery and what other interventions may already have been administered. As such, other implementation approaches, such as algorithms, protocols, checklists or activation of haemorrhage codes, might be more appropriate.[70 71] Defining the optimal implementation approach for early detection and first response management of PPH during and after caesarean birth still remains to be completed. Conducting the necessary research to answer this question should be an immediate next step.

In addition, efforts should be pursued to agree on standardised approaches for the management of refractory PPH during and after caesarean birth. Importantly, these standardised approaches should encompass both the specific interventions used to manage refractory PPH, appropriate fluid and blood product management protocols and the implementation strategies to support their uptake and sustainability. Standardised approaches will need to be applicable to various settings, including those with limited access to laboratories, crossmatched blood and blood products, expensive devices and medical

specialists. This consensus focuses mainly on obstetric interventions, although haemodynamic resuscitation and obstetric measures to stop haemorrhage should be applied simultaneously. Recommendations for haemostatic resuscitation, including haemodynamic, coagulopathy, transfusion and intraoperative cell salvage,[72] will be part of the forthcoming WHO/FIGO/ICM (International Confederation of Midwives) consolidated PPH guidelines in 2024 (Althabe, personal communication).

## CONCLUSION

This expert consensus proposes strategies for early detection and first response to PPH during and after caesarean birth. Future research should determine how best to implement these strategies and evaluate the effectiveness of the proposed implementation approach. Such research should be conducted soon, so that the approaches and interventions proposed here can rapidly be operationalised and institutionalised to contribute to the global efforts to reduce maternal death and disability.

**Author affiliations**
[1]Institute for Clinical Effectiveness and Health Policy, Buenos Aires, Argentina
[2]Department of Maternal & Child Health, University of North Carolina at Chapel Hill Gillings School of Global Public Health, Chapel Hill, North Carolina, USA
[3]Department of Obstetrics & Gynaecology, Muhimbili University of Health and Allied Sciences, Dar es Salaam, United Republic of Tanzania
[4]Department of Global Public Health, Karolinska Institute, Stockholm, Sweden
[5]Maternidad Martin, Secretaría de Salud Pública de la Municipalidad de Rosario, Rosario, Santa Fe, Argentina
[6]Centro de Estudios de Estado y Sociedad, Buenos Aires, Argentina
[7]Department of Obstetrics & Gynaecology, St George's Hospital, London, UK
[8]Department of Public Health, Faculty of Medicine, Universidad de Buenos Aires, Buenos Aires, Argentina
[9]Stanford University School of Medicine, Stanford, California, USA
[10]Obstetrical Perinatal and Pediatric Epidemiology Research team, Centre for Research in Statistics and Epidemiology (CRESS) Université Paris Cité INSERM, Paris, France
[11]Research in Childbirth and Health, University of Central Lancashire, Preston, UK
[12]THRIVE Centre, School of Heath and Community Studies, University of Central Lancashire, Preston, UK
[13]CEPED, Université Paris Cité, IRD, INSERM, Paris, France
[14]Departamento de Ginecología y Obstetricia, Fundación Valle del Lili, Cali, Colombia
[15]Facultad de Ciencias de la Salud, Universidad Icesi, Cali, Colombia
[16]Technical Leadership & Innovations Office, Jhpiego/USA, Baltimore, Maryland, USA
[17]Department of Obstetrics and Gynaecology, University of Cape Town, Rondebosch, South Africa
[18]Department of Obstetrics and Gynaecology, Aminu Kano Teaching Hospital, Kano, Nigeria
[19]Africa Center of Excellence for Population Health and Policy, Bayero University Kano, Kano, Nigeria
[20]Obstetrics and Gynecology, Hung Vuong Hospital, Ho Chi Minh, Viet Nam
[21]Department of Obstetrics and Gynaecology, University of Botswana, Gaborone, Botswana
[22]Effective Care Research Unit, University of the Witwatersrand, Johannesburg and Walter Sisulu University, Mthatha, South Africa
[23]Burnet Institute, Melbourne, Victoria, Australia
[24]Amherst College, Amherst, Massachusetts, USA
[25]Department of Epidemiology, Prince of Songkla University, Hat Yai, Thailand
[26]Department of Obstetrics & Gynecology, Prince of Songkla University, Hat Yai, Thailand
[27]Department of Obstetrics and Gynaecology, Khon Kaen University, Khon Kaen, Thailand
[28]Department of Obstetrics & Gynecology—Maternal Fetal Medicine, Stanford University, Stanford, California, USA
[29]California Maternal Quality Care Collaborative, Standford, California, USA
[30]Liverpool School of Tropical Medicine, Nairobi, Kenya
[31]Institute of Metabolism and Systems Research, University of Birmingham, Birmingham, UK
[32]Department of Obstetrics and Gynaecology, Singleton Hospital, Swansea Bay University Health Board, Swansea, UK
[33]Department of Obstetrics & Gynecology, Ain Shams University Faculty of Medicine, Cairo, Egypt
[34]Department of Obstetrics and Gynaecology, Gaia/ Espinho Local Health Unit, Vila Nova de Gaia, Portugal
[35]RISE-HEALTH - CINTESIS—Center for Health Technology and Services Research, University of Porto, Porto, Portugal
[36]Faculty of Medicine, University of Porto, Porto, Portugal
[37]Department of Mother and Child Health Research, Institute for Clinical Effectiveness and Health Policy, Buenos Aires, Argentina
[38]Department of Women's and Children's Health, University of Liverpool, Liverpool, UK
[39]Department of Delivery, Tu Du Hospital, Ho Chi Minh City, Viet Nam
[40]University of Nairobi Department of Obstetrics and Gynecology, Nairobi, Kenya
[41]Woman and Reproduction Health Unit at Maternal Health at the Latin American Center of Perinatology (CLAP/WR), Pan American Health Organization, Montevideo, District of Columbia, USA
[42]Department of Obstetrics and Gynecology, School of Medicine, Universidad de la República Uruguay, Montevideo, Uruguay
[43]Women's Health, Global Surgery Foundation, Washington, District of Columbia, USA
[44]Liverpool Women's Hospital, Liverpool, UK
[45]UNDP-UNFPA-UNICEF-WHO-World Bank Special Programme of Research, Development and Research Training in Human Reproduction (HRP), Department of Sexual and Reproductive Health and Research, WHO, Geneva, Switzerland
[46]Bixby Center for Global reproductive Health, University of California San Francisco, San Francisco, California, USA

**Acknowledgements** We would like to acknowledge the contribution of Bill & Melinda Gates Foundation officers for conceiving the research question and supporting the conduct of this study; the contribution of Zhitong Yu in preparing the consensus building process and the administrative support of Chastine Sebolino and Maria Harmitton Oliveto.

**Contributors** SM, FA, AC, IG and OTO conceived the idea. VP developed the protocol with input from SM and FA. SM, FA, CRW and VP coordinated the project with input from IG and AC. FM, AL, VO, AB, CRW and VP conducted the systematic review. OTO, IG, AC, EA, SA, AN, FMAA, BC, CD-T, SD, AD, MFE, CE, SF, HSG, GJH, CH, TL, PL, EM, IN, JM, DTH, TP, ZPQ, CS, JV, AW, FA, SM, CRW and VP participated in the consensus and provided expert input at various stages of the project. Data were analysed by CRW and VP with input from SM and FA. SM, FA, CRW and VP wrote the manuscript with input from MW and all coauthors.

**Funding** This study was funded by the Bill & Melinda Gates Foundation (INV-001393) through a grant to the University of Birmingham, UK, and the UNDP/UNFPA/UNICEF/WHO/World Bank Special Programme of Research, Development and Research Training in Human Reproduction (HRP), a cosponsored programme executed by WHO.

**Disclaimer** The author is a staff member of the World Health Organization. The author alone is responsible for the views expressed in this publication and they do not necessarily represent the views, decisions or policies of the World Health Organization. This manuscript represents the views of the authors and not the views of WHO or the UNDP/UNFPA/UNICEF/WHO/World Bank Special Programme of Research, Development and Research Training in Human Reproduction (HRP).

**Competing interests** Disclosure forms provided by the authors are available with the full text of this article.

**Patient and public involvement** Patients and/or the public were not involved in the design, or conduct, or reporting, or dissemination plans of this research.

**Patient consent for publication** Not applicable.

**Ethics approval** Given the consultative nature of the exercise which involved consenting experts in their work capacity, ethical clearance was not required. Participants gave informed consent to participate in the study before taking part.

**Provenance and peer review** Not commissioned; externally peer reviewed.

**Data availability statement** Data are available on reasonable request. Extensive data are presented in the supplementary materials. Given the small sample size, datasets will be available on reasonable request.

**ORCID iDs**
Verónica Pingray http://orcid.org/0000-0002-7889-2825
Caitlin R Williams http://orcid.org/0000-0002-4925-869X
Brendan Carvalho http://orcid.org/0000-0002-4919-4542
Tippawan Liabsuetrakul http://orcid.org/0000-0001-7687-5629
Francis G Muriithi http://orcid.org/0000-0002-2314-5611
Ashraf Fawzy Nabhan http://orcid.org/0000-0003-4572-2210
Thuan N Q Phan http://orcid.org/0000-0002-7166-8345

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
