## [Reviewer comments · BMJ Open]

ARTICLE DETAILS

TITLE (PROVISIONAL)	Strategies for optimising early detection and obstetric first response management of postpartum haemorrhage at caesarean birth: A modified Delphi-based international expert consensus
AUTHORS	Pingray, Verónica; Williams, Caitlin R.; Al-beity, Fadhlun M; Abalos, Edgardo; Arulkumaran, Sabaratnam; Blumenfeld, Alejandro; Carvalho, B; Deneux-Tharoux, Catherine; Downe, Soo; Dumont, Alexandre; Escobar, Maria Fernanda; Evans, Cherrie; Fawcus, Sue; Galadanci, HS; Hoang, Diem-Tuyet; Hofmeyr, G Justus; Homer, Caroline; Lewis, Ayodele; Liabsuetrakul, Tippawan; Lumbiganon, Pisake; Main, Elliot; Maua, Judith; MURIITHI, FRANCIS GITHAE; Nabhan, Ashraf; NUNES, Inês; Ortega, Vanesa; Phan, Thuan; Qureshi, Zahida; Sosa, Claudio; Varallo, John; Weeks, Andrew; Widmer, Mariana; Oladapo, Olufemi; Gallos, Ioannis; Coomarasamy, Arri; Miller, Suellen; Althabe, Fernando

VERSION 1 – REVIEW

REVIEWER	Brownfoot, Fiona University of Melbourne, Obstetrics and Gynaecology
REVIEW RETURNED	16-Oct-2023

GENERAL COMMENTS	Strategies for optimising early detection and first response management of postpartum haemorrhage at caesarean birth: A modified Delphi-based international expert consensus. I would like to congratulate the authors for an excellent systematic review and three stage modified-Delphi expert consensus examining postpartum haemorrhage at caesarean section. It simplifies the diagnosis of PPH and clearly outlines first stages of managing haemorrhage at caesarean section. The text was enjoyable to read. The tables clearly outline description and management including the various medications and procedures to stop bleeding. I have a few questions:  1. I wonder if the authors might comment on blood group and whether this should be considered? 2. I wonder about comment on massive transfusion and activation of a haemorrhage code? Or this may be outside the scope of the review. 3. Many surgical approaches were described however I didn't see hysterectomy mentioned. Again, this may be outside the scope of this review. Excellent manuscript.
---

REVIEWER	Ducloy-Bouthors, Anne-Sophie
-----------------	------------------------------

	Centre Hospitalier Regional Universitaire de Montpellier Pole Mere et enfant
REVIEW RETURNED	01-Nov-2023

GENERAL COMMENTS	The authors report an international experts'panel process to improve the early detection and first step management of caesarean (CS)_induced postpartum hemorrhage (PPH). After a systematic review, they submitted a three-stage modified-Delphi consensus to 22 international experts. The authors must be thanked for their scientific approach to clarify the definition of PPH in CS. Some points need to be reviewed regarding the resuscitation procedures (hemodynamic, cell salvage, coagulopathy, transfusion). The rigorous patient blood management (PBM) as soon as PPH begins is a key of success for woman'health. The authors comment themselves the need for an upgraded PBM part of the delphi process following the FIGO and US and French integration of the crucial detection and therapeutic actions. If they cannot review their manuscript, because the delphi course is closed, please notice in the abstract and conclusion that the PBM part of the detection and management are not updated. Authors : Did the authors all contribute to the conception of the delphi questions or did they also answer to the questions ? Is there a way to identify which author have a daily clinical function in direct contact with bleeding patients and which are epidemiologist or health care organization maker ? For exemple by adding the woman hospital name and city to the clinical practitioners' affiliation. Anesthesiologists and intensivists should have been associated to the approach as most of the patient blood management (PBM) procedure are elusive, not up-to-date and not precise enough to guide correctly clinicians in CS-induced PPH PBM. Key words : please add tranexamic acid to the Key words list Introduction : The problematic of caesarean delivery induced PPH impact on maternal mortality and morbidity is well described. Questions and answers about caesarean section induced-PPH detection and first step management P12 I57: to measure the vaginal bleeding during caesarean section and appreciate closely its progression, It is helpful to install a collector bag at the buttock of the patient between the espacements of the operating table (AS Ducloy-Bouthors et al Rev. Méd. Périnat. (2018) 10:200-202). This have been instaured as a routine practice in women hospitals as a part of quantitative detection of PPH in CS. P12 chapter first response intraoperative phase. Only 3 causes are considered : trauma tissue and tone. Thrombin is not
---

	considered. However coagulopathy may be a serious cause of PPH primary to any atony or any bleeding as suggested in recent new insights papers (Bell SF, et al. Int J Obstet Anesth. 2021;47:102983; Karlsson, O., A. J et al Int J Obstet Anesth 2014 ; 23 : 10-7 ; Collins PW, et al. Blood 2014 ; 124 : 1727-36; Oliver J. et al Int J Obstet Anaesth 2022 ;51 :103573 ; Lucy De Lloyd et al J Thromb Haemost. 2022;:1–18 ; Hofer S et al. Eur J Anaesthesiol. 2023 ; 40(1): 29–38 ; Munoz et al Blood Transfus. 2019 Mar; 17(2): 112–13). PPH can also occur in patients with inherited bleeding disorders that may be revealed by placental bed incoercible bleeding. ð Please add to this detection and first management step that a special attention must be carried on coagulation process and detection of acute obstetric coagulopathy (AOC).. Blood and bleeding fluidity can be observed by the clinicians; AOC can be assessed by a delocalised first and early step analysis : dry tube’s coagulation time or point of care fibrinogen device or viscoelastic test confirmed secondarily by the lab tests In cases of acute obstetric coagulopathy, tranexamic acid is not able to reverse fibrinogenolysis and fibrinogen function drastic decrease (Lucy De Lloyd et al J Thromb Haemost. 2022;:1–18; Ducloy-Bouthors AS et al Br J Anaesth. 2016;116(5):641-8. Ducloy-Bouthors AS et al Br J Anaesth. déc 2022;129(6):937-45).Fibrinogen decrease must be supplemented early, parallely to the antifibrinolytic action of tranexamic acid by plasma or cryoprecipitate or fibrinogen concentrates. The plasma fibrinogen threshold to be reached is 2g/L. ð please add the correction of hypofibrinogenemia as a first step management if drastic fibrinogen decrease < 2g/L appears primary (maniotic fluid embolism, placental abruptio, placenta spectrum disease, prolonged fetal death) or in the course of high flow massive haemorrhage . p13 l15 to l22 The present chapter reinforce my questions about clinicians as authors and contributors. “Unclear” cannot be written by a competent clinician. Hemodynamic resuscitation is a very well and clearly codified challenge for anesthesiologists and intensivists. Objectives, monitoring devices and therapeutic tools are not different in PPH compared to other hemorrhagic conditions. Please give the pregnant women the right to the best resuscitation procedures. Patient blood management procedures includes hemodynamix resuscitation, are well described and guidelines are written (Hofer S et al. Eur J Anaesthesiol. 2023 ; 40(1): 29–38 ; Munoz et al Blood Transfus. 2019 Mar; 17(2): 112–13, Gestion du capital sanguin en obstétrique HAS 2022; Shaylor R et al Anesth Analg. 2017 Jan; 124(1): 216–232.). Detection of hemodynamic failure can be made simply via the shock index follow up. The current objectives are as follows : mean arterial blood pressure > 65 mmHg or systolic BP > 80 mmHg and/or a heart rate <100bpm and/or urinary flow > 120mL/3 hrs. Cardiac output can be followed up by non invasive monitoring. Hemodynamic maintenance must be initiated early by cristalloids and inotrope drugs and transfusion. Massive transfusion protocols must be initiated as soon as possible in cases of high flow PPH.
--	---

	Access to blood products must be anticipated as a part of patient blood management protocols. P13 please add : In cases of massive high flow hemorrhage, cell salvage should be installed as early as possible for transfusion if available (Hofer S et al. Eur J Anaesthesiol. 2023 ; 40(1): 29–38 ; Munoz et al Blood Transfus. 2019 Mar; 17(2): 112–13, Gestion du capital sanguin en obstétrique HAS 2022). P14 I26 Because of TXA elimination in high flow bleeding, TXA dose should be renewed proportionally of the drug waste (gilliot S et al pharmaceutics 2022;14-578). Postoperative P15 I7 : counting and weighting pads. Scales are available in every labour wards for babyweight measurement and should be used to monitor severe PPH P15 I27 as in intraoperative phase, the supposed lack of evidence about hemodynamic status monitoring and resuscitation let anesthesiologists and midwives manage at their own feeling: “local protocols” only because obstetric experts involved in the present scientific approach never are not specialists of hemodynamic resuscitation. Please give to pregnant women the right to the best resuscitation procedures they are allowed to wait from our teams.
--	--

VERSION 1 – AUTHOR RESPONSE

Reviewer	Reviewer Comment	Response
Reviewer #1	Would like to congratulate the authors for an excellent systematic review and three stage modified-Delphi expert consensus examining postpartum haemorrhage at caesarean section. It simplifies the diagnosis of PPH and clearly outlines first stages of managing haemorrhage at caesarean section. The text was enjoyable to read. The tables clearly outline description and management including the various medications and procedures to stop bleeding.	Thank you for this comment. It is indeed an issue that requires evidence generation.
Reviewer #1	I wonder if the authors might comment on blood group and whether this should be considered?	We thank the reviewer for this comment. It is indeed a relevant issue; however, it was not within the scope of our consensus, which focused mainly on obstetric interventions. We took this opportunity to make it clear by revising the manuscript (page 2, line 5, page 6, lines 7 and 20, page 7, line 29, page 9, box 1). In addition, we added the description of good surgical practices in Supplementary File S6.
Reviewer #1	I wonder about comment on massive transfusion and activation of a haemorrhage code? Or this may be outside the scope of the review.	Thank you. As mentioned above, the focus was obstetric intervention, and transfusion was not within the scope of our consensus. We clarified this by inserting the following paragraph (page 7, lines 22-25)., "Although most cases of PPH are controlled by the simultaneous application of obstetric interventions and haemostatic support, this consensus focused mainly on obstetric interventions and not haemostatic resuscitation and treatment of anaemia and coagulopathy." Regarding the haemorrhage code, the reviewer mentions a successful implementation strategy that we have not suggested in the section we dedicated to implementation in our manuscript. Following the reviewer's suggestion, we have edited the text "It is unclear whether a bundle is the most appropriate implementation approach for PPH during and after caesarean birth. Bundles require a set of interventions to be administered together, but

Reviewer	Reviewer Comment	Response
		the administration of some of the clinical interventions outlined here may depend on what occurs during surgery and what other interventions may already have been administered. As such, other implementation approaches, such as algorithms, protocols, or checklists, or activation of haemorrhage codes might be more appropriate (Page 20, lines 21-26)
Reviewer #1	Many surgical approaches were described however I didn't see hysterectomy mentioned. Again, this may be outside the scope of this review.	That is an excellent point, and we are glad that the reviewer identified that the manuscript is not clear enough regarding excluding hysterectomy. Because the review and consensus aimed to identify and agree on methods of detection and first response to PPH during or after CS, the investigators assumed that the first response would not include non-conservative interventions, which should be considered for PPH refractory to the first response. The systematic review had conservative interventions as an eligibility criterion. Thank you for identifying this flaw in the manuscript. We have clarified this point on page 5, line 24, page 6, line 20 and Box 1.
Reviewer #2	The authors must be thanked for their scientific approach to clarify the definition of PPH in CS.	Thank you for the comment. We are glad to hear that the reviewer considers our approach to be a valuable scientific contribution.
Reviewer #2	Some points need to be reviewed regarding the resuscitation procedures (hemodynamic, cell salvage, coagulopathy, transfusion). The rigorous patient blood management (PBM) as soon as PPH begins is a key of success for woman'health. The authors comment themselves the need for an upgraded PBM part of the delphi process following the FIGO and US and French integration of the crucial detection and therapeutic actions. If they cannot review their manuscript, because the delphi course is closed, please notice in the abstract and conclusion that the PBM part of the detection and management are not updated.	Thanks to the reviewer for the comment. Undoubtedly a very important point. Haemodynamic resuscitation was outside of the scope of our study. We acknowledge that further clarity was required in our manuscript, so we inserted the following paragraph."Although most cases of PPH are controlled by the simultaneous application of obstetric interventions and haemostatic support, this consensus focused mainly on obstetric interventions but not haemostatic resuscitation and treatment of anaemia and coagulopathy." (page 7, lines 22-24). In addition, we systematically mentioned that the consensus was targeting mainly obstetric interventions (page 2, line 5, page 6, lines 7 and 20, page 7, line 29, page 9, box 1). Finally, in the discussion section, we added: "Standardised approaches will need to be applicable to a variety of settings, including those with limited access to laboratories, crossmatched blood and blood products, expensive devices, and medical specialists. This consensus

Reviewer	Reviewer Comment	Response
		focuses mainly on obstetric interventions, although haemodynamic resuscitation and obstetric measures to stop haemorrhage should be applied simultaneously. Recommendations for haemostatic resuscitation, including hemodynamic, coagulopathy, transfusion, and intraoperative cell salvage (72), will be part of the forthcoming WHO/FIGO/ICM consolidated PPH guidelines in 2024 (F Althabe personal communication).” (Page 21, lines 3-11)
Reviewer #2	Authors : Did the authors all contribute to the conception of the delphi questions or did they also answer to the questions ? Is there a way to identify which author have a daily clinical function in direct contact with bleeding patients and which are epidemiologist or health care organisation maker ? For exemple by adding the woman hospital name and city to the clinical practitioners' affiliation. Anesthesiologists and intensivists should have been associated to the approach as most of the patient blood management (PBM) procedure are elusive, not up-to-date and not precise enough to guide correctly clincians in CS-induced PPH PBM.	Thank you for this comment. Authors either conceived, coordinated or participated in the Delphi consensus-building process. This is described in the section "CONTRIBUTION TO AUTHORSHIP". The experts who participated all have clinical backgrounds and conduct clinical research in maternity hospitals. 14 of the 22 experts are practising obstetricians, intensivists or anesthesiologists, warranting the representation of clinicians' perspectives. Experts in anesthesiology and maternal critical care were included in the panel and participated throughout all three study stages to provide their perspectives. As we mentioned before, and although we acknowledge the relevance of providing guidance on PBM, patient blood management was outside the scope of our study. Please see the responses to previous comments.
Reviewer #2	Key words : please add tranexamic acid to the Key words list	Thank you for this valuable comment. We now include Tranexamic acid in the keywords list (page 3).
Reviewer #2	P12 I57: to measure the vaginal bleeding during caesarean section and appreciate closely its progression, It is helpful to install a collector bag at the buttock of the patient between the espacements of the operating table (AS Ducloy-Bouthors et al Rev. Méd. Périnat. (2018) 10:200-202). This have been instaured	Thank you very much for this important point. This theme emerged from the discussions, but perhaps we did not describe it clearly enough. We edited the manuscript to make it as straightforward as possible (page 12, lines 54-56 and page 15, Box 3).

Reviewer	Reviewer Comment	Response
	as a routine practice in women hospitals as a part of quantitative detection of PPH in CS.	
Reviewer #2	P12 chapter first response intraoperative phase. Only 3 causes are considered : trauma tissue and tone. Thrombin is not considered. However coagulopathy may be a serious cause of PPH primary to any atony or any bleeding as suggested in recent new insights papers (Bell SF, et al. Int J Obstet Anesth. 2021;47:102983; Karlsson, O., A. J et al Int J Obstet Anesth 2014 ; 23 : 10-7 ; Collins PW, et al. Blood 2014 ; 124 : 1727-36; Oliver J. et al Int J Obstet Anaesth 2022 ;51 :103573 ; Lucy De Lloyd et al J Thromb Haemost. 2022;;1–18 ; Hofer S et al. Eur J Anaesthesiol. 2023 ; 40(1): 29–38 ; Munoz et al Blood Transfus. 2019 Mar; 17(2): 112–13). PPH can also occur in patients with inherited bleeding disorders that may be revealed by placental bed incoercible bleeding.	Thank you for this comment. While this was a very important point which we described in the systematic review protocol, it is not clearly described in the manuscript. Thank you very much for identifying this weakness in the first version of the manuscript. Both the systematic review and the Delphi have the scope to identify interventions for the first response. The scope of the study includes treatment options for women who develop primary PPH during or after caesarean birth at the time of initiating treatment, when the aetiology is suspected to be uterine atony, traumatic PPH or unknown. Treatments for managing women diagnosed with coagulopathy, placenta praevia, and placenta accreta were not included, as treatments are usually specific for each aetiology. We edited the manuscript to clarify this in the objectives and methodology, the scope and type of target population for the interventions identified, and agreed to provide the first response to CS PPH (page 5, lines 23-25; page 6, lines 9-13; page 7 lines 16-22).
Reviewer #2	Please add to this detection and first management step that a special attention must be carried on coagulation process and detection of acute obstetric coagulopathy (AOC).. Blood and bleeding fluidity can be observed by the clinicians; AOC can be assessed by a delocalised first and early step analysis : dry tube's coagulation time or point of care fibrinogen device or viscoelastic test confirmed secondarily by the lab tests	We agree with the reviewer that detecting and managing coagulopathies and specifically AOC is critical. However, it was outside of the scope of our study. We acknowledge that further clarity was required in our manuscript. For this reason, we revised the manuscript and clarified that some aetiologies were outside the scope of our project (page 5, lines 23-25; page 6, lines 9-13; page 7 lines 16-22). In addition, we added the following sentences for clarification in the discussion section: “Importantly, these standardised approaches should encompass both the specific interventions used to manage refractory PPH, appropriate fluid and blood product management protocols, and the implementation strategies to support their uptake and sustainability. Standardised approaches will need to be applicable to a variety of settings, including those with limited access to laboratories, crossmatched blood and blood products, expensive devices, and medical specialists. This consensus focuses mainly on obstetric interventions, although haemodynamic

Reviewer	Reviewer Comment	Response
		resuscitation and obstetric measures to stop haemorrhage should be applied simultaneously. Recommendations for haemostatic resuscitation, including hemodynamic, coagulopathy, transfusion, and intraoperative cell salvage (72), will be part of the forthcoming WHO/FIGO/ICM consolidated PPH guidelines in 2024 (F Althabe personal communication).” Page 21, 3-11
Reviewer #2	In cases of acute obstetric coagulopathy, tranexamic acid is not able to reverse fibrinolysis and fibrinogen function drastic decrease (Lucy De Lloyd et al J Thromb Haemost. 2022;:1–18; Ducloy-Bouthors AS et al Br J Anaesth. 2016;116(5):641-8. Ducloy-Bouthors AS et al Br J Anaesth. déc 2022;129(6):937-45).Fibrinogen decrease must be supplemented early, parallelly to the antifibrinolytic action of tranexamic acid by plasma or cryoprecipitate or fibrinogen concentrates. The plasma fibrinogen threshold to be reached is 2g/L.	Thank you for pointing out this issue. Since the treatment of AOC is out of scope, this topic was not discussed among the experts and, therefore, is not addressed in the summary of the discussions; therefore the specific detection methods and treatment are not described.
Reviewer #2	please add the correction of hypofibrinogenemia as a first step management if drastic fibrinogen decrease < 2g/L appears primary (maniotic fluid embolism, placental abruptio, placenta spectrum disease, prolonged fetal death) or in the course of high flow massive haemorrhage. p13 l15 to l22 The present chapter reinforce my questions about clinicians as authors and contributors. "Unclear" cannot be written by a competent clinician. Hemodynamic resuscitation is a very well and clearly codified challenge for anesthesiologists and intensivists. Objectives, monitoring devices and therapeutic tools are not different in PPH compared to other hemorrhagic conditions. Please give the pregnant women the right to the best resuscitation procedures. Patient blood management procedures includes hemodynamix resuscitation, are well described and guidelines are written (Hofer S et al. Eur J Anaesthesiol. 2023 ; 40(1): 29–38 ; Munoz et al Blood Transfus. 2019 Mar; 17(2): 112–13, Gestion du capital	We also believe that pregnant women deserve the very best management; however, as we previously noted, the study mainly focused in obstetric interventions for first response management, and we have edited the manuscript accordingly for clarity. We further clarify that the agreed interventions may not be appropriate for cases that require an individualised approach due to the quantity and rapidity of blood loss (Page 14 lines 26-27). Although the consensus mainly focused on obstetric first-response management, hemodynamic resuscitation emerged during the expert discussion, and some experts noted that providing guidance on amounts of fluids was too case-specific. However, others stressed that inexperienced clinicians needed concrete guidance to avoid adding excessive fluids and inducing fluid overload. Although this type of guidance is beyond the scope of this study, it is a relevant issue that should be addressed. We added this

Reviewer	Reviewer Comment	Response
	sanguin en obstétrique HAS 2022; Shaylor R et al Anesth Analg. 2017 Jan; 124(1): 216–232.). Detection of hemodynamic failure can be made simply via the shock index follow up. The current objectives are as follows : mean arterial blood pressure > 65 mmHg or systolic BP > 80 mmHg and/or a heart rate <100bpm and/or urinary flow > 120mL/3 hrs. Cardiac output can be followed up by non invasive monitoring. Hemodynamic maintenance must be initiated early by cristalloids and inotrope drugs and transfusion. Massive transfusion protocols must be initiated as soon as possible in cases of high flow PPH. Access to blood products must be anticipated as a part of patient blood management protocols.	explanation in page 14 lines 11-24 and in the discussion section (page 21, lines 3-11). In addition, the experts acknowledged that providing guidance on haemodynamic parameters cut off points for postoperative thresholds to trigger treatment will help clinicians act more quickly. Several experts raised the possibility of using the Shock Index (heart rate divided by systolic blood pressure; OSI) as a clinical decision support tool to simplify the decision of when to act, given that it has been used in some settings, including low-resource settings. We revised the manuscript to add this clarification (page 16, lines 11-21).
	P13 please add : In cases of massive high flow hemorrhage, cell salvage should be installed as early as possible for transfusion if available (Hofer S et al. Eur J Anaesthesiol. 2023 ; 40(1): 29–38 ; Munoz et al Blood Transfus. 2019 Mar; 17(2): 112–13, Gestion du capital sanguin en obstétrique HAS 2022).	Thanks for the comment. Due to the focus on obstetric interventions, the systematic review did not include the use of cell salvage. Consequently, it wasn't discussed among experts. Nonetheless, we recognize its relevance and acknowledge that the forthcoming WHO/FIGO/ICM consolidated PPH guidelines in 2024 will address this theme by describing "This consensus focuses mainly on obstetric interventions, although haemodynamic resuscitation and obstetric measures to stop haemorrhage should be applied simultaneously. Recommendations for haemostatic resuscitation, including hemodynamic, coagulopathy, transfusion, and intraoperative cell salvage (72), will be part of the forthcoming WHO/FIGO/ICM consolidated PPH guidelines in 2024 (F Althabe personal communication)." (Page 21, lines 3-10). In addition, we clarified that the agreed interventions may not be appropriate for cases that require an individualised approach due to the amount and

Reviewer	Reviewer Comment	Response
		rapidity of blood loss (this would include massive high-flow haemorrhage) (Page 14 lines 26-27).
	P14 l26 Because of TXA elimination in high flow bleeding, TXA dose should be renewed proportionally of the drug waste (gilliot S et al pharmaceuticals 2022;14-578).	Thanks for the observation. The PPH guidelines included in the systematic review do not mention any recommendation to adjust the dose or timing according high-flow bleeding. Furthermore, the issue did not occur during the discussion with the experts. We prefer not to introduce potentially controversial issues supported by single studies.
	Postoperative P15 l7 : counting and weighting pads. Scales are available in every labour wards for babyweight measurement and should be used to monitor severe PPH	Thanks for the suggestions. We have edited the manuscript to describe that counting and weighting pads should be considered whenever possible (Page 16, lines 1-4).
	P15 l27 as in intraoperative phase, the supposed lack of evidence about hemodynamic status monitoring and resuscitation let anesthesiologists and midwives manage at their own feeling: "local protocols" only because obstetric experts involved in the present scientific approach never are not specialists of hemodynamic resuscitation. Please give to pregnant women the right to the best resuscitation procedures they are allowed to wait from our teams.	Thank you for highlighting this important issue. It was vital not to mislead the reader and note that this consensus-building process focuses mainly on obstetric interventions. As stated above, we have tried to clarify this throughout the manuscript. Thank you very much.